# Impact of the COVID-19 Pandemic on Mental Health among Patients with Chronic Ocular Conditions

Soumaya Bouhout [1], Mélanie Hébert [2] , Weronika Jakubowska [1] , Laurence Jaworski [1,3], Ellen E. Freeman [4,5] and Marie-Josée Aubin [1,6,7,*]

[1] Department of Ophthalmology, Université de Montréal, Montréal, QC H3T 1J4, Canada
[2] Department of Ophthalmology, Université Laval, Quebec City, QC G1V 0A6, Canada
[3] University Ophthalmology Center, Centre Hospitalier de l'Université de Montréal (CHUM), Montréal, QC H2X 3E4, Canada
[4] School of Epidemiology and Public Health, University of Ottawa, Ottawa, ON K1G 5Z3, Canada
[5] Ottawa Hospital Research Institute, Ottawa, ON K1Y 4E9, Canada
[6] University Ophthalmology Center, Centre Intégré Universitaire de Santé et de Services Sociaux de l'Est-de-l'Île-de-Montréal–Hôpital Maisonneuve-Rosemont, Montréal, QC H1T 2M4, Canada
[7] Department of Social and Preventive Medicine, School of Public Health, Université de Montréal, Montréal, QC H3T 1J4, Canada
* Correspondence: marie-josee.aubin@umontreal.ca

**Abstract:** The COVID-19 pandemic had significant impacts on the mental and visual health of patients. This cross-sectional, survey-based, multicentric study evaluates the state of mental and visual health among patients with chronic ocular diseases such as glaucoma, neovascular age-related macular degeneration, diabetic retinopathy, or chronic uveitis during the lockdown period of the COVID-19 pandemic. Mental health was assessed using three questionnaires: the Patient Health Questionnaire-9 (PHQ-9), the Impact of Event Scale-Revised (IES-R), and the National Eye Institute Visual Function Questionnaire-25 (VFQ-25). A total of 145 patients completed the questionnaires. The PHQ-9 showed that most respondents ($n = 89$, 61%) had none or minimal depressive symptoms, while 31 (21%) had mild depressive symptoms, 19 (13%) had moderate depressive symptoms, 5 (3%) had moderately severe depressive symptoms, and 1 (1%) had severe depressive symptoms. Regarding stress surrounding the pandemic, the median IES-R showed mild distress in 16 (11%), moderate distress in 7 (5%), and severe distress in 4 (3%). The COVID-19 pandemic lockdowns had a negative impact on patients' mental health with close to 20% of the patients reporting at least moderately depressive symptoms and 19% reporting at least mildly distressful symptoms.

**Keywords:** coronavirus 2019; ophthalmology; mental health

## 1. Introduction

The 2019 coronavirus (COVID-19) pandemic emerged as a global crisis, affecting every aspect of human life [1,2]. Beyond its public health implications, the pandemic has had a profound impact on mental wellbeing, exacerbating pre-existing mental health conditions and amplifying the development of underlying mental health conditions [3]. Numerous studies have highlighted the detrimental effects of the COVID-19 pandemic on mental health [3–7]. The COVID-19 pandemic has been associated with increased feelings of loneliness, hopelessness, anxiety, frustration, and despair [3,4]. The implementation of quarantine measures, lockdowns, social distancing protocols, and widespread fear and uncertainty have collectively contributed to increased rates of anxiety, depression, and psychological distress [4–6].

During the pandemic, numerous changes were implemented in patient care and health services to adhere to global and local public health guidelines [4]. In order to reduce and prevent the spread of COVID-19, only cases categorized as urgent or semi-urgent were evaluated and treated, leading to a reduction in non-essential activities. Many countries

experienced dramatic reductions in ophthalmology consultations and elective surgery to less than 38% and 11% of normal volumes, respectively [8]. A majority of ophthalmologists, including up to 80% in some countries, did not perform elective surgeries during the peak of the pandemic [8]. Some institutions have used telemedicine as an alternative for the follow-up of patients to reduce in-person assessments [9]. However, the sudden disruption of routine eye care services and deferred care resulting from reduced clinical volume and elective procedure postponement have heightened the risk of ocular complications and vision loss, especially among populations at risk of chronic ocular diseases [8].

Individuals with chronic diseases represent a vulnerable population due to their health comorbidities and reliance on healthcare services. Chronic ocular diseases encompass a broad spectrum of conditions affecting visual health and have also been correlated to mental health disorders such as depression. For example, diabetic retinopathy is associated with all-cause and cardiovascular mortality, depression, eating disorders, and several forms of cognitive impairment [10]. Patients with chronic ocular conditions such as glaucoma, age-related macular degeneration (AMD), proliferative diabetic retinopathy (PDR), and chronic uveitis may be at higher risk of experiencing reduced visual function and quality of life related to the burden of their visual condition. Patients with these chronic ocular conditions require regular follow-ups to monitor the progression of their chronic diseases and prevent severe vision loss [11]. In addition, these patients were also at greater risk of COVID-19 complications due to their age, comorbidities, and immunosuppression [1,12]. These patients had to face many challenges during the pandemic as they had to balance the risk of contagion against potential visual loss caused by the progression of their ocular disease. The pandemic-induced restrictions, including limited access to healthcare services, delayed follow-up visits and treatments, and reduced social support networks, may have further compounded the psychosocial impact of their ocular condition during the pandemic.

Therefore, this study aims to explore the specific challenges faced by patients with chronic ocular disease to better understand the interplay between their visual health and mental wellbeing in the context of the COVID-19 pandemic. This study seeks to shed light on the mental health repercussions and psychological distress faced by vulnerable patients at risk of vision loss. Understanding these challenges will help guide patient care and the development of targeted interventions and support systems to mitigate adverse mental health outcomes in the setting of a global health crisis.

## 2. Materials and Methods

### 2.1. Study Design and Population

This cross-sectional study assessed the mental and visual health of patients during the COVID-19 pandemic using validated questionnaires. Patients at greater risk for loss to follow-ups were targeted, using the following criteria for the selection of patients: adult patients (≥18 years old) capable of providing informed consent, at least 3 visits in ophthalmology per year since February 2018, having a scheduled follow-up in March 2020, and a confirmed diagnosis of at least one ocular disease, including chronic glaucoma, wet AMD requiring anti-vascular endothelial growth factor (VEGF) injections, PDR, or chronic uveitis treated with systemic immunosuppressants or corticosteroid drops (e.g., prednisolone 1%, dexamethasone 0.1%, difluprednate 0.05%) at least twice daily. Patients were selected based on the presence of chronic ocular conditions regardless of mental health history. Patients from two university tertiary care centers were included (Centre Universitaire d'Ophtalmologie–Hôpital Maisonneuve-Rosemont and Centre Hospitalier de l'Université de Montréal). Data collection included age, sex, systemic health conditions, underlying ocular condition, visual acuity, and intraocular pressure of both eyes at the final follow-up before the initial pandemic lockdown on 13 March 2020. Patients were contacted over the phone to complete the questionnaires between June 2021 and December 2021. To be included in the analysis, patients needed to have completed at least one questionnaire. This study was approved by the ethics board of each center.

## 2.2. Mental and Visual Health Questionnaires

Patients were contacted via phone, and informed consent was obtained verbally. To obtain informed consent, patients were first contacted, and if they agreed to be part of the study, the questionnaire was given 48 h later in order to give them the time to withdraw from this study. To assess the mental and visual health of patients, a combination of validated questionnaires was used based on the respondent's preferred language (i.e., French or English). The Patient Health Questionnaire-9 (PHQ-9) [13] was used to assess depressive symptoms, the Impact of Event Scale-Revised (IES-R) [14] was used to assess the impact of the pandemic on respondents' stress and sleep, and visual health was assessed using the National Eye Institute Visual Function Questionnaire-25 (VFQ-25) [15–17].

The PHQ-9 has been recognized as a reliable and valid tool for screening major depressive disorder [13]. The PHQ-9 score ranges from 0 to 27 and includes nine questions regarding symptoms of depression that are graded as 0 (not at all), 1 (several days), 2 (more than half the days), and 3 (nearly every day). The total score can be categorized as representing none or minimal depressive symptoms from 0 to 4, mild symptoms from 5 to 9, moderate symptoms from 10 to 14, moderately severe symptoms from 15 to 19, and severe symptoms when greater than 20. Another item of the questionnaire allows respondents to characterize how difficult these symptoms have rendered daily functioning, and impairment is deemed present when they respond, "very difficult" or "extremely difficult".

The IES-R is a valid tool to assess the response to a traumatic event [18]. Scores range from 0 to 88 and are categorized as follows: normal (0–23), mild distress (24–32), moderate distress (33–36), and severe distress (≥37) [19]. Scores of ≥33 represent the best cut-off for probable posttraumatic stress disorder (PTSD) [20], while at ≥37, this score becomes high enough to suppress immunity even 10 years after the event [21].

The VFQ-25 is a comprehensive general and visual health questionnaire that assesses different aspects via subscales for general health, general vision, ocular pain, near activities, distance activities, social functioning, mental health, role difficulties, driving, color vision, peripheral vision, as well as a composite score [17]. Each is graded on a scale of 0 to 100 using the mean of the provided answers for each subscale item, with higher scores denoting better health. Optional additional questions are available to improve the reliability of the questionnaire. These were offered to the patients but remained optional.

## 2.3. Statistical Analysis

Study data were collected and managed using REDCap electronic data capture tools. REDCap (Research Electronic Data Capture) is a secure, web-based software platform designed to support data capture for research studies [22]. Data are presented as mean ± standard deviation (SD) for continuous, normally distributed variables, as median [first quartile, third quartile] for continuous, non-normally distributed variables, and as frequencies (percentages) for categorical variables. Characteristics and variables were compared between groups of differing severity in mental health symptoms (e.g., age, sex, ethnicity, and ocular disease) using independent Student's *t*-test, analysis of variance (ANOVA), Mann–Whitney U test, or Kruskal–Wallis test as appropriate for continuous variables and chi-square analysis for categorical variables. Shapiro–Wilk test and Q-Q plots with 95% confidence intervals were used to test for normality of distribution in continuous variables. Spearman's rho was used to evaluate correlation between different questionnaire scores. A Spearman's rho of 0.00 to 0.29 was deemed a negligible correlation, 0.30 to 0.49 a weak correlation, 0.50 to 0.69 a moderately strong correlation, 0.70 to 0.89 a strong correlation, and 0.9 to 1.0 a very strong correlation [23]. Box-and-whiskers plots were used to illustrate the differences in visual function using VHQ-25 scoring by differing severity in mental health symptoms to verify whether a worse visual function had an impact on mental health symptoms during the pandemic.

Statistical analyses were performed using R for Windows (version 3.6.3; R Foundation for Statistical Computing) and IBM SPSS Statistics for Windows (version 27.0; IBM Corp., Armonk, NY, USA). Analyses were conducted at a significance level of 0.05.

## 3. Results

### 3.1. Patient Baseline Characteristics

A total of 360 patients were eligible for participation, of which 145 patients answered the questionnaires (40%). One patient did answer two of three questionnaires (VFQ-25 and PHQ-9), and all the remaining patients answered all three. Baseline characteristics are presented in Table 1. The median [Q1, Q3] age was 69 [59, 75] years, with 75 (52%) female patients. A majority of patients (89%) were Caucasian, and many (>40%) had metabolic disorder comorbidities (e.g., hypertension, dyslipidemia, diabetes). The distribution of ocular conditions between chronic glaucoma, wet AMD, PDR, and chronic uveitis is well distributed. The median visual acuity in logMAR was 0.18 for both the right and left eyes. In our cohort, 6.4% of patients reported having a diagnosed psychiatric condition.

**Table 1.** Baseline characteristics and demographics of patients who answered the questionnaires.

| Characteristic | Total Cohort, *n* = 145 |
|---|---|
| Age, median [Q1, Q3] | 69 [59, 75] |
| Female sex, *n* (%) | 75.52% |
| Ethnicity, *n* (%) | |
|    Caucasian | 88.60% |
|    Asian | 6.40% |
|    Black | 2.10% |
|    Afro-Caribbean | 1.10% |
|    Native Indian | 0.00% |
|    Hispanic | 3.20% |
|    Unknown | 42.29% |
|    Other | 4.30% |
| Ocular disease | |
|    Glaucoma | 37.26% |
|    Age-related macular degeneration | 36.25% |
|    Proliferative diabetic retinopathy | 39.27% |
|    Chronic uveitis | 44.30% |
|    Combination of disease | 12.8% |
| Visual acuity, *logMAR* | |
|    Right eye, median (IQR) | 0.18 [0.10, 0.36] |
|    Left eye | 0.18 [0.06, 0.60] |
| Intraocular pressure (mmHg) | |
|    Right eye | 16 [12, 18] |
|    Left eye | 15 [12, 18] |
| Systemic medical conditions | |
|    No medical condition | 14.10% |
|    Chronic obstructive pulmonary disease or asthma | 10.70% |
|    Interstitial lung disease | 1.10% |
|    Diabetes | 48.33% |
|    Morbid obesit (body mass index > 40) | 1.10% |
|    Hypertension | 63.43% |
|    Dyslipidemia | 49.34% |
|    Cardiovascular disease (CAD, CHF) | 15.10% |
|    Chronic kidney disease or end-stage renal disease | 7.50% |
|    Cancer | 9.60% |
|    Inflammatory bowel disease | 6.40% |
|    Liver disease | 3.20% |
|    Chronic neurological or neuromuscular disease | 5.30% |
|    Psychiatric condition (e.g., BP) | 6.40% |
|    Rheumatologic disease | 12.80% |
|    Unknown | 24.17% |
|    Other | 14.10% |

Data presented as number, % for categorical values accounting for missing data and median [Q1, Q3] for continuous variable. BP = bipolar disorder; CAD = coronary artery disease; CHF = congestive heart failure.

### 3.2. Mental Health Questionnaire Responses

For depressive symptoms, the median [Q1, Q3] PHQ-9 score was 3 [1, 8] (range: 0–46). Most respondents (*n* = 89, 61%) had none or minimal depressive symptoms, while 31 (21%) had mild depressive symptoms, 19 (13%) had moderate depressive symptoms, 5 (3%) had moderately severe depressive symptoms, and 1 (1%) had severe depressive symptoms. Of these, 15 (10%) had an impaired daily functioning. There were no significant differences in the presence of depressive symptoms, overall depressive symptoms score, nor in the severity category of depressive symptoms (*p* > 0.05) between the principal disease category (i.e., presence of depressive symptoms in chronic glaucoma, *n* = 9, 29%; wet AMD requiring anti-VEGF injections, *n* = 18, 53%; PDR, *n* = 14, 38%; or chronic uveitis, *n* = 15, 35%; *p* = 0.22). Furthermore, as illustrated in Figure 1, there was a statistically significant moderate correlation between the PHQ-9 and the VFQ-25 scores, with lower scores of the VFQ-25 questionnaire associated with more severe symptoms of depression (ρ = −0.434, *p* < 0.001). There was a lower score in patients with the presence of depressive symptoms compared to those without (71 [56, 86] vs. 87 [79, 93], *p* < 0.001).

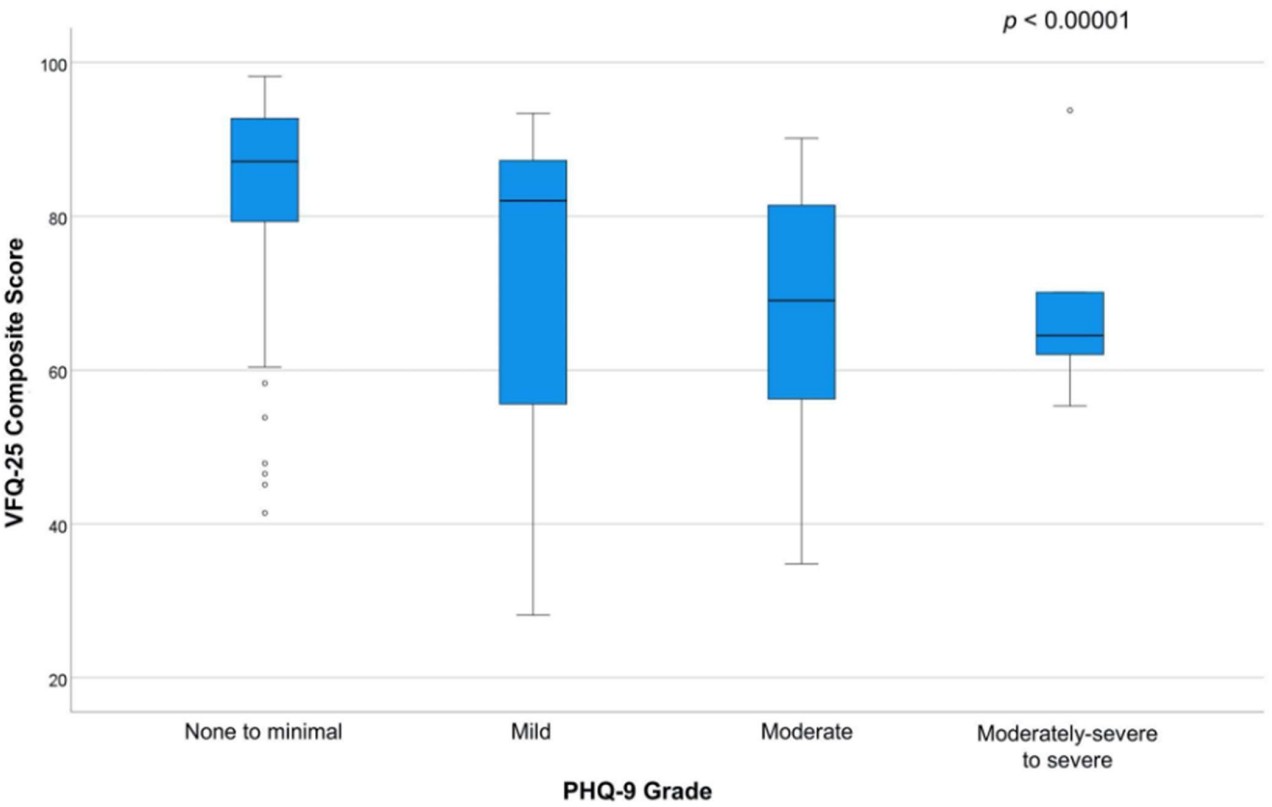

**Figure 1.** Box-and-whiskers plots illustrating the results of the National Eye Institute Visual Function Questionnaire-25 (VFQ-25) by severity of the Patient Health Questionnaire-9 (PHQ-9).

Regarding stress surrounding the pandemic, the median IES-R was 11 [5, 18] (range: 0–46). No significant distress was found in 117 (81%), mild distress in 16 (11%), moderate distress in 7 (5%), and severe distress in 4 (3%). There were 11 (8%) who had scores ≥33, indicating a probable diagnosis of PTSD [18], and 4 (3%) had scores ≥37 compatible with the suppression of immunity [19]. IES-R scores did not differ for overall score nor for other distress categories (*p* > 0.05) between disease categories (i.e., presence of distress symptoms in chronic glaucoma *n* = 10, 32%, wet AMD requiring anti-VEGF injections *n* = 5, 15%, PDR *n* = 6, 16%, or chronic uveitis *n* = 6, 14%; *p* = 0.19). Furthermore, as illustrated in Figure 2, there was a statistically significant mild correlation between the VFQ-25 and the IES-R scores, with lower VFQ-25 scores associated with worse symptoms of distress (ρ = −0.311,

*p* < 0.001). There was also a lower VFQ-25 score in patients with the presence of distress symptoms compared to those without (74 [58, 88] vs. 85 [72, 91]; *p* = 0.02).

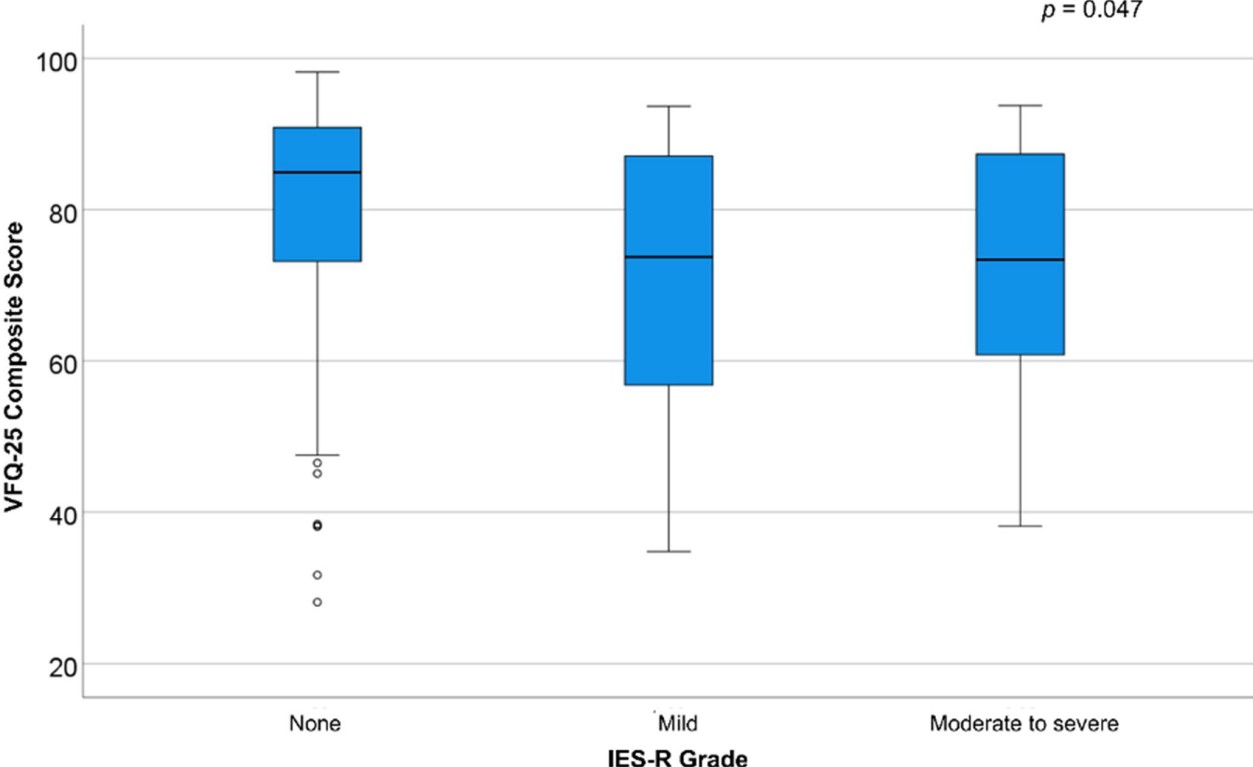

**Figure 2.** Box-and-whiskers plots illustrating the results of the National Eye Institute Visual Function Questionnaire-25 (VFQ-25) by severity of the Impact of Event Scale-Revised (IES-R).

For general and visual health, the median composite VFQ-25 was 84 [70, 91]. A total of 101 patients (70%) answered the optional items, which did not differ by principal disease category (*p* > 0.05). Each subscale is presented in Table 2. Differences regarding mental health, near and distant activities, driving, and composite scores were found between disease categories and were significantly lower in the AMD group, most likely due to worse baseline visual acuity (Supplementary Tables S1 and S2).

**Table 2.** Results of the National Eye Institute Visual Function Questionnaire-25 subscale among 145 patients.

| Subscale * | Median [Q1, Q3] |
|---|---|
| General health | 60 [48, 75] |
| General vision | 65 [55, 80] |
| Ocular pain | 88 [75, 100] |
| Near activities | 79 [58, 96] |
| Distance activities | 92 [75, 100] |
| Vision specific: | |
| ○　　Social functioning | 100 [92, 100] |
| ○　　Mental health | 80 [50, 94] |
| ○　　Role difficulties | 88 [63, 100] |
| ○　　Dependency | 100 [67, 100] |
| Driving | 83 [67, 100] |
| Color Vision | 100 [100, 100] |
| Peripheral Vision | 100 [75, 100] |
| Composite Score | 84 [70, 91] |

* Each is graded on a scale of 0 to 100 using the mean of the provided answers for each subscale item, with higher scores denoting better health.

## 4. Discussion

In this multicenter cross-sectional study, we reported results of validated questionnaires regarding mental and visual health in patients with diverse chronic ocular diseases during the first wave of the COVID-19 pandemic. We showed that close to one in five patients had moderate symptoms of depression and signs of distress surrounding the pandemic. This is more than twice the previously reported rates in the general population [24]. Our study has highlighted a vulnerable population who needs regular follow-up and/or treatments and how the lockdown affected patients with chronic ocular conditions. It gives insight into future policies in case of a future lockdown.

During the first wave of the COVID-19 pandemic, worldwide lockdowns were put in place to face the rapid spread of the virus. Following the guidelines of multiple institutions, including the American Academy of Ophthalmology and the Canadian Ophthalmology Society, regular follow-ups and surgeries were delayed, and only urgent cases were seen in person [25–28]. These policies have led to a significant decrease in clinic visits (up to 70–79%) [29], affecting patients with chronic diseases requiring regular follow-ups, such as glaucoma, chronic uveitis on immunosuppressants and/or chronic utilization of topical steroid drops, diabetic retinopathy, and AMD requiring injections [30–33]. Furthermore, decreased attendance by patients due to fear of contracting the virus has also been reported [32,34]. Regular follow-up is key to preventing permanent vision loss from these chronic conditions [28,35–37]. Telemedicine in ophthalmology has been used for follow-ups, and most patients have appreciated this modality [38]. However, telemedicine has major limitations for the follow-up and treatments of ocular diseases such as anti-VEGF injections or glaucoma or uveitis clinical visits, which require direct assessment and examination of patients with specialized ophthalmic equipment that is not accessible in the setting of telemedicine (slit lamp, tonometry, OCT imaging, visual field testing, and others). Emerging reports have already shown an increase in sight-threatening complications, such as submacular hemorrhages in AMD during the COVID-19 lockdown [39]. In addition, vision loss can be a contributing factor to depressive symptoms, cognitive decline, and morbidity, especially in the elderly population [40–42].

Moreover, previous studies have shown the negative impacts of the pandemic lockdown among healthcare professionals, the general population, and patients living with chronic diseases such as multiple sclerosis [6–8,43,44]. One study has recently investigated the prevalence of depression among AMD patients during the pandemic using the World Health Organization Five Well-Being Index (WHO-5), equivalent to the PHQ-9 score used in this study, and showed the pandemic may have had a negative impact on mental health [45]. Furthermore, in the United Kingdom, Ting et al. have shown a significant negative psychosocial impact on individuals affected by eye diseases during pandemic lockdowns related to limited access to treatment and care [46]. Additionally, 39% of patients reported difficulty coping with lockdowns due to their ophthalmic condition. However, previous studies neither included validated questionnaires nor assessed visual health. Our study provides insight into the psychological impacts of COVID-19 on patients affected by chronic ocular conditions using validated questionnaires. Eye care professionals should be aware of the potential negative impacts of the pandemic on patient mental and visual health to promote better patient care.

In terms of visual impact, we reported similar VFQ-25 scores to the literature across the different pathologies [47–50]. Average visual acuity was statistically significantly different across the groups (Supplementary Material Table S1) and seemed to be lower in retinal pathologies, especially AMD. The latter could explain the lower VFQ-25 score in the AMD group. In addition, the findings demonstrate an association between lower visual function scores and heightened symptoms of depression and distress related to the pandemic. This highlights the vulnerability of patients with low vision, particularly in the aftermath of a traumatic event such as the pandemic.

This study highlights the importance of one aspect that is often overlooked: mental health among patients with chronic ocular conditions. Ophthalmologists closely follow

some of their patients in the clinic, particularly those with chronic ocular diseases such as diabetic retinopathy, requiring visits as often as every few months. These frequent interactions present exceptional opportunities for evaluating various facets of patient health and care, including mental health, especially in times of adversity like the pandemic. Given that patients with lower vision and those with chronic ocular conditions are at higher risk of experiencing symptoms of psychological distress or depression, they could be screened for mental health symptoms within the setting of ophthalmology clinics. This could be conducted by educating healthcare to enhance their awareness of mental health symptoms among vulnerable populations. In addition, nurses that work alongside ophthalmologists could perform questionnaires such as PHQ-9 to screen for depressive symptoms. Mental health screening could be integrated into the clinic workflow by having nurses administer screening questionnaires while patients are in the waiting room. Based on their scores, patients with depressive symptoms could further be referred to mental health resources such as a support phone line or be evaluated by a professional team. In the event of a future lockdown or pandemic, the integration of mental health screening and assessment tools into clinical practices could improve the well-being and quality of life among patients.

One of the limitations of this study is a moderate response rate, which can be attributed partly to the way in which the questionnaires were administered. To respect the standards of the Institutional Review Board, patients needed to be contacted via telephone at least twice: first to explain the study goals and again to obtain informed consent, allowing the patient time to reflect on their choice to participate. This requirement to reach the patient twice made participant retention more difficult, and some who may have consented to this study initially would be unreachable for the administration of the questionnaire. Likewise, given that mental health may still be considered taboo in certain patient populations, this could also impact the rate of patient participation in this study. This could have two contrary effects: either dissuading patients who have mental health symptoms from participating for fear of being labeled with these symptoms or dissuading patients who do not have mental health symptoms because they do not feel compelled by the subject to answer the survey.

The implementation of various public health policies related to the pandemic has impacted the mental health of the general population. We believe that the timing of the surveys in relation to the pandemic, which corresponds to the fourth wave of COVID-19 in Canada (between June 2021 and December 2021), may have affected the IES-R scores of patients as public health policies were changing. One of the major limitations is not having a baseline response rate prior to the pandemic and no available data regarding patients' previous mental health history. This study is also subject to possible response bias and recalls bias because questionnaires were administered more than a year after the beginning of the pandemic. Finally, this study did not investigate the patients' perspective regarding their ocular condition or their appreciation of the impact the pandemic has had on their eye care. Additionally, to the best of our knowledge, no studies have looked into the impact of telemedicine on the mental health of patients followed in ophthalmology during the COVID-19 pandemic.

## 5. Conclusions

This study highlights the negative psychological impacts of the COVID-19 pandemic on patients with chronic ocular conditions, with close to one in five patients reporting experiencing depression and/or some level of distress. This study also found an overall association between vision-related quality of life and mental health parameters regardless of disease category. Consequently, mental health assessment and screening using questionnaires such as PHQ-9 should be considered for patients with chronic ocular disease, especially in the setting of a pandemic or any other public health crisis. Additional research is needed to develop and implement resources for mental health adapted for patients with ocular comorbidities.

**Supplementary Materials:** The following supporting information can be downloaded at: https://www.mdpi.com/article/10.3390/vision7030049/s1, Table S1: Baseline characteristics of the cohort by principal disease category; Table S2: National Eye Institute Visual Function Questionnaire-25 subscale answers by principal disease category.

**Author Contributions:** Conceptualization, S.B., M.H., W.J., L.J., E.E.F. and M.-J.A.; methodology, S.B., M.H., W.J., L.J., E.E.F. and M.-J.A.; validation, S.B., M.H., W.J., L.J., E.E.F. and M.-J.A.; formal analysis, S.B., M.H. and W.J.; investigation, S.B., M.H. and W.J.; Resources, M.-J.A.; data curation, S.B. and M.H.; writing—original draft, S.B., M.H. and W.J.; writing—review and editing, S.B., M.H., W.J., L.J., E.E.F. and M.-J.A.; visualization, S.B., M.H. and W.J.; supervision, E.E.F. and M.-J.A.; project administration, M.-J.A.; funding acquisition, M.H. and M.-J.A. All authors have read and agreed to the published version of the manuscript.

**Funding:** This research was funded by the National and International Networking Program of the Vision Health Research Network and the philanthropic funds of the Vice-Dean of Research and Development of the Université de Montréal.

**Institutional Review Board Statement:** The study was conducted in accordance with the Declaration of Helsinki and approved by the Institutional Review Board of the Centre intégré universitaire de santé et de services sociaux de l'Est-de-l'Île-de-MontréaL (MP-12-2021-2299; 24 August 2020).

**Informed Consent Statement:** Informed consent was obtained from all subjects involved in this study.

**Data Availability Statement:** The dataset used in this current study is available from the corresponding author upon reasonable request.

**Acknowledgments:** The authors wish to acknowledge all the participants who took the time to answer surveys for this study.

**Conflicts of Interest:** The authors declare no conflict of interest. The funders had no role in the design of the study; in the collection, analyses, or interpretation of data; in the writing of the manuscript; or in the decision to publish the results.

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
