# Peer review of "Impact of the COVID-19 Pandemic on Mental Health among Patients with Chronic Ocular Conditions"

_2411-5150, 2023_

Round 1

Reviewer 1 Report

1.       The title is better to be modified, some words like “ophthalmology patients” make no sense; please change it with suitable words such as “referral patients to the ophthalmic clinic” or you can remove it with no extra rewording.

2.       The study rational is not stated clearly, why did you select patients with diabetic retinopathy, age related macular degeneration, uveitis and etc. for examination. Because of the chronic diseases is not a good reason for eligibility. Please more clarify.

3.       When you claimed that patients with older ages are at a higher risk of COVID- 19, please refer to the suitable study as a citation.

4.       The variety of confounding variables were interacted with the final results for instance visual impairment in different diseases serveries could be influential in deteriorating of the vision related quality of life outcomes!!! How did you manage it?

5.       Please kindly report the mean value of the visual acuity in both better and worse eyes.

6.       Please add this fact that you entered healthy patients regarding the psychological conditions to your study.

7.       What were your definitions for mild, moderate term that extensively applied throughout the text for all the variables even for defining the correlations?

8.       Please repeat the statistical analysis based on the cut points considered as an acceptable values and report percentage with 95% confidence interval for each factor.

No comments

Author Response

  1. The title is better to be modified, some words like “ophthalmology patients” make no sense; please change it with suitable words such as “referral patients to the ophthalmic clinic” or you can remove it with no extra rewording.

    Thank you for your comment the title has been changed to the following: Impact of the COVID-19 Pandemic on Mental Health among Patients with Chronic Ocular Conditions

  2. The study rational is not stated clearly, why did you select patients with diabetic retinopathy, age related macular degeneration, uveitis and etc. for examination. Because of the chronic diseases is not a good reason for eligibility. Please more clarify.

We have selected patients with various common chronic ocular conditions such as diabetic retinopathy, AMD, glaucoma, and uveitis because they represent some of the most frequent conditions followed regularly in ophthalmology clinics and are at risk of progression and visual loss. In addition, some of these ocular conditions such as diabetic retinopathy have been associated with mental health disorders.

“Individuals with chronic diseases represent a vulnerable population due to their health comorbidities and reliance on health care services. Chronic ocular diseases encompass a broad spectrum of conditions affecting visual health, and have also been correlated to mental health disorders such as depression. For example, diabetic retinopathy is associated with all-cause and cardio-vascular mortality, depression, eating disorders and several forms of cognitive impairment [10]. Patients with chronic ocular conditions such as glaucoma, age-related macular degeneration (AMD), proliferative diabetic retinopathy (PDR), and chronic uveitis may be at higher risk of experiencing reduced visual function and quality of life related to the burden of their visual condition. Patients with these chronic ocular conditions require regular follow-ups to monitor the progression of their chronic diseases and prevent severe vision loss [11].” (Line 60-70)

  1. When you claimed that patients with older ages are at a higher risk of COVID- 19, please refer to the suitable study as a citation.

We have added an approriate reference supporting reasons why older people are at higher risk of COVID-19 and of having complications such as increased comorbitites (Line 71-72).
New reference added #12: Mueller, A.L.; McNamara, M.S.; Sinclair, D.A. Why does COVID-19 disproportionately affect older people? Aging (Albany NY). 2020, 12(10), 9959-9981. doi: 10.18632/aging.103344.

  1. The variety of confounding variables were interacted with the final results for instance visual impairment in different diseases serveries could be influential in deteriorating of the vision related quality of life outcomes!!! How did you manage it?

In the analyses, we have included all the comparisons between disease severities for each mental health metric which were not significantly different. Furthermore, the differences in vision-related quality of life by subscale of the NEI-VFQ-25 are presented in Supplementary Table 2. Given that there were no differences in mental health metrics by disease category however, the overall association between vision-related quality of life and mental health parameters are presented in the figures.

  1. Please kindly report the mean value of the visual acuity in both better and worse eyes.

Median [Q1, Q3] is presented in the revised manuscript and in the Supplementary Table 2 by disease category. Better eye: 0.10 [0.00, 0.30] and worse eye: 0.30 [0.18, 0.88].

  1. Please add this fact that you entered healthy patients regarding the psychological conditions to your study. 

Thank you for your comment. The following sentence has been added to the manuscript: “Patients were selected based on the presence of chronic ocular condition regardlesss of mental health history.” (line 95-97)

  1. 7.      What were your definitions for mild, moderate term that extensively applied throughout the text for all the variables even for defining the correlations?

The following definition was used to define correlations with Spearman rho’s testing:

“A Spearman’s rho of 0.00 to 0.29 was deemed a negligible correlation, 0.30 to 0.49 a weak correlation, 0.50 to 0.69 a moderately strong correlation, 0.70 to 0.89 a strong correlation, and 0.9 to 1.0 a very strong correlation” (line 155-168)

We added the following new reference #23 : Hinkle DE, Wiersma W, Jurs SG. Applied statistics for the behavioral sciences. 3rd ed. Boston: Houghton Mifflin; 1994. 706 p.

  1.  Please repeat the statistical analysis based on the cut points considered as an acceptable values and report percentage with 95% confidence interval for each factor.

The statistical analyses were repeated using both VFQ by presence or absence of depressive or distress symptoms respectively, as well as disease category by presence or absence of depressive or distress symptoms using the normal cutoffs. Given that these are absolute proportions, there were no 95% confidence intervals to be produced.

Reviewer 2 Report

This is a very interesting manuscript and suitable for publication in this journal.

I only have two comments.

- Explain in more detail the titles of the figures

- Explain how these meetings can help in future clinical practice

Author Response

  1. Explain in more detail the titles of the figures

Thank you for your comments the following modifications have been done to the manuscript:

Table 1. Baseline characteristics and demographics of patients who answered the questionnaires.
Table 2.  Results of the National Eye Institute Visual Function Questionnaire-25 subscale  among 145 patients.

Figure 1. Relation Correlation between the results of the National Eye Institute Visual Function Questionnaire-25 (VFQ-25) and the Patient Health Questionnaire-9 (PHQ-9). 

Figure 2. Relation Correlation between the results of the National Eye Institute Visual Function Questionnaire-25 (VFQ-25) and the Impact of Event Scale-Revised (IES-R).

  1. Explain how these meetings can help in future clinical practice
    Thank you for your comment. The following paragraph has been edited in the manuscript :

“Ophthalmologists follow closely some of their patients in clinic, particularly those with chronic ocular dieases such as diabetic retinopathy requiring visits as often as every few months. These frequent interactions present exceptional opportunities for evaluating various facets of patient health and care including mental health, especially in times of adversity like the pandemic. Given that patients with lower vision and those with chronic ocular conditions are at higher risk of experiencing symptoms of psychological distress or depression, they could be screened for mental health symptoms within the setting of ophthalmology clinics. This could be done by educating healthcare to enhance awareness of mental health symptoms among uvlnerable populations. In addition, nurses that work alongside ophthalmologists could perform questionnaires such as PHQ-9 to screen for depressive symptoms. Mental health screening could be integrated in the clinic workflow by having nurses administer screening questionnaires while patients are in the wait room. Based on their scores, patients with depressive symptoms could further be referred to mental health ressources such as a support phone line or be evalued by a profesionnal team. In the event of a future lockdown or pandemic, the integration of mental-health screening and assessment tools into clinical practices could improve the well-being and enhance quality of life among vulnerable patients.“ (lines 343-359).

Round 2

Reviewer 1 Report

All the comments have been considered in the text. 

English editing is recommended to be conducted by a native English editor. 

Author Response

English revision was conducted by a native English editor.  A version with all the track changes has been attached. 
